# Assessing Behavioral Disorders with SDQ in Very Preterm Children at 5 Years of Age in LIFT Cohort

**DOI:** 10.3390/children10071191

**Published:** 2023-07-10

**Authors:** Marine Robert de Saint Vincent, Valérie Rouger, Jean Christophe Rozé, Cyril Flamant, Jean-Baptiste Muller

**Affiliations:** 1Department of Neonatal Medicine, University Hospital of Nantes, 44000 Nantes, France; 2Loire Infant Follow-Up Team (LIFT) Network, 44000 Nantes, France; valerie.bureau@chu-nantes.fr; 3National Institute of Health and Medical Research, CIC 1413, University Hospital of Nantes, 44000 Nantes, France

**Keywords:** preterm, functional development outcome, behavior, SDQ

## Abstract

Background: Preterm-born children are at risk of behavioral disorders and the systematic assessment of these disorders remains a challenge. Questions remain about the accuracy of self-reported parent questionnaires and the real everyday life behavior of the child. Aim: To evaluate the association between SDQ reported by parents in the preterm and behavioral difficulties in the everyday school life environment reported by teacher. Methods: All children born before 33 weeks and who followed-up in the LIFT (Loire Infant Follow-up team) network were included. The Strengths and Difficulties Parental Questionnaire (SDQ), completed at 5 years, was used to check for behavioral difficulties and identified three groups: “normal”, “borderline” and “abnormal”. Then, the SDQ results were compared to the Global School Adaptation Score (GSA) at 5 years. Results: Out of the 1825 children followed in the cohort at the age of 5, 1397 questionnaires were analyzed. A total of 11.1% of children had an abnormal score, and 9.7% had a borderline score. Male gender and a lower birth weight z-score were significantly associated with the “abnormal SDQ” group. There is a significant relationship between the probability of being in the “abnormal SDQ” group at 5 years and with difficulty in global school adaptation at 5 years, as well as an SDQ borderline score in the preterm (*p* < 0.001). Conclusions: SDQ abnormal and borderline scores are associated with behavioral difficulties in the classroom and everyday life behavior. In preterm children, one should be alerted even by a borderline SDQ score.

## 1. Background

Advances in neonatology have increased the survival rate of very preterm infants [1]. However, prolonged resuscitative care affects brain development at a critical period. This increases the risk of long-term neurological sequelae [2]. The main sequelae are motor, sensory and cognitive, and manifest themselves gradually over the years [3]. For these reasons, specific monitoring would be appropriate in order to detect developmental difficulties at an early stage and to set up relevant care.

Many studies have shown an association between preterm birth and the occurrence of behavioral problems [4,5,6,7,8], using different questionnaires or scores such as the Achenbach questionnaire (Child Behavior Checklist), FTF (Five To Fifteen questionnaire), BRIEF (Behavior Rating Inventory of Executive Function) [9] and others adapted to children of different ages. All describe a strong link between preterm birth and behavioral problems before 5 years of age. Some studies have focused on behavioral problems in young, preschool-aged children [10], which may disappear as the child grows [11]. Preterm birth and environmental factors related to prematurity, such as socioeconomic status, play a role in behavior problems development [12].

The Strengths and Difficulties Questionnaire (SDQ) is a questionnaire that was developed in 1997 by Robert Goodman [13] and seeks to assess behavioral problems in children. It was designed to facilitate the detection of these disorders, accessible to parents and teachers, while remaining reliable. Several studies have already demonstrated the ability of the SDQ to assess children’s behavior with respect to their mental health in particularly. For instance, SDQ is reliable in detecting behavioral problems in children with deafness [14] or epilepsy [15].

The evaluation of classroom everyday-life behavior functioning can be assessed with the GSA questionnaire. Designed by Paul Guimard and Agnès Florin [16], the GSA was originally defined as a tool for teachers to assess children’s abilities and behavior in the classroom. The questionnaire has been tested as a useful tool for the early detection of learning disabilities and behavioral problems in preschool children. It has been demonstrated that the GSA questionnaire is also highly consistent with full-scale IQ scores [17].

Child behavioral disorders can be related to the family environment. These psychoaffective disorders do not take place in all lively place of the child, unlike constitutional disorders. To ensure that the SDQ parental reports explore constitutional disorders, we compared the SDQ to the GSA. The objective of the present study is to investigate the association between SDQ parental reports and child behavior in the classroom with GSA reported by the teacher of 5-year-olds in the LIFT preterm cohort of children.

## 2. Methods

### 2.1. Subjects

This study included all surviving infants born between April 2008 and January 2013 at a gestational age of less than 33 weeks who were enrolled in the LIFT cohort. Written consent was obtained from the parents of each child before inclusion. The LIFT network includes 24 maternity clinics and three hospitals with neonatal intensive care units (Nantes, Angers, Le Mans), and follows up with vulnerable children up to the age of 7. The aim of this network is to detect early clinical problems and to provide patient-specific care. Standard follow-ups are carried out at 3, 9, 12, 18 and 24 months and at 3, 4, 5 and 7 years. Written consent was obtained for each patient before inclusion in the study and the cohort was registered with the French Data Protection Authority for clinical research (CNIL, No. 851117).

### 2.2. Study Design

At the age of 5, the follow-up is based on an interview with the parents, a full clinical examination including clinical tests to assess fine and gross motor abilities, oral language, memory and questionnaires filled in before the consultation by the parents and/or the child’s teachers.

The GSA (Global School Adaptation) questionnaire assesses the classroom everyday-life behavior of children [16]. Parents received and forwarded it to the child’s teacher. A total of 11 questions concern the child’s behavior in the classroom, 3 questions assess linguistic competence, 6 questions investigate non-verbal abilities such as memory, arithmetic, manual skills, and the last question asks the teacher to provide their prognosis for the child’s future adaptation to school life.

The SDQ (Strengths and Difficulties Questionnaire) is used to identify behavioral problems in children. Robert Goodman described it in 1997, based on the 1996 Rutter scale [13], which is faster to complete than the CBCL (Child Behavior Checklist) described by Achenbach in 1991 [18]. The questionnaire explores five distinct dimensions: conduct disorders, emotional disorders, peer relationships, hyperactivity and prosocial score. The SDQ is applicable to children aged 4 to 16 years and can be completed by parents, teachers or the child themself. The French translation of the SDQ was tested on a sample of French school-aged children to ensure cross-cultural validity [19]. The final version of the SDQ-Fra consists of 25 questions that correspond to the five dimensions. A total of 10 questions are generally considered strengths, 14 are difficulties, and one question is neutral. For each question, 3 answers are possible: “Not true”, “Sometimes or somewhat true”, and “Very true”, each corresponding to a score of 0, 1, or 2, respectively, depending on whether the question is aimed at a strength or a difficulty. The score for each dimension is obtained by adding the scores of the 5 questions exploring the domain, with 10 corresponding to the highest stage of the dimension. By combining the scores for each of the difficulties, a total difficulty score is calculated, scored from 0 to 40, and includes conduct, emotional, peer relationship, and hyperactivity disorders. Thresholds for each score were set so that all children scoring above the 90th percentile were considered “abnormal”, all those between the 80th and 90th percentile as “borderline”, and all others “normal”.

Perinatal and socioeconomic data from the LIFT cohort registry were collected for the children included in the study. These data included gestational age, sex, birth weight, the existence of bronchopulmonary dysplasia, the presence of brain lesions (occurrence of an IVH of grade 3 or 4, or periventricular leukomalacia identified with ultrasound monitoring during neonatal hospitalization), the insured status of the Universal Medical Insurance, the relationship status of the parents (couple or not) and the highest professional level of the mother.

### 2.3. Statistical Analysis

Results are reported as means and standard deviations for continuous variables and as numbers of subjects and percentages for categorical variables. The significance level was determined by *p* < 0.05 for all analyses. An SDQ was “abnormal” if the total difficulty score was between 17 and 40, “borderline” between 14 and 16, and normal between 0 and 13. A GSA score under 48 is associated with a risk of school-based learning difficulties. To investigate the role of neonatal and socioeconomic data on the occurrence of each disorder, we performed a multiple regression. Statistical analyses were performed using SPSS software.

## 3. Results

A total of 2244 children were enrolled in the LIFT cohort during the study period, of whom 1825 (81.3%) had a visit at 5 years of age, 1397 (62.2%) were assessed with the SDQ and 1235 (55.0%) with both the SDQ and GSA questionnaires (Figure 1).

Characteristics of the study population compared to the population for which questionnaires were not received or not completed are presented in Table 1. There was no significant difference between the two populations with respect to gestational age, birth weight, sex and bronchopulmonary dysplasia rate (*p* > 0.05 for each of these data). There were differences in socioeconomic conditions with more patients receiving CMUc (free complementary health insurance), indicating a lower socioeconomic level in the not-included population as well as a tendency for more frequent severe neurologic complications in this group.

Table 2 describes the SDQ total difficulty score according to the perinatal data. Respectively, 11.2% and 9.8% of children had an abnormal and a borderline score. Regarding the perinatal data, significant differences were found for the Z-scores of sex and birth weight. There was a non-significant tendency for CMU recipients in the borderline and abnormal groups.

Children in the LIFT network with a normal total difficulty score had a mean GSA of 49.8 (±7.9), those with a borderline score had a mean GSA of 46.1 (±7.8) and those with an abnormal score had a mean GSA of 43.5 (±8.19) (*p* < 0.001).

The GSA score for the different SDQ categories according to the gender of the child is presented in Figure 2.

Of the 1235 children with both questionnaires completed, 979 had a normal SDQ score, and 67.2% of these children also had a normal GSA. A total of 134 children had an abnormal SDQ difficulty score, and of these children, 70.9% also had an abnormal GSA. A total of 122 children had a borderline SDQ score, and of these children, 57.4% had an abnormal GSA (chi-square 28.64, *p* < 0.001).

In the logistic regression model including the variables of gestational age, birth weight Z-score, gender, bronchopulmonary dysplasia, brain damage, CMU and breastfeeding, only birth weight Z-score (*p* = 0.011) and sex (*p* < 0.001) were significantly related to non-optimal SDQ (defined as borderline or abnormal total difficulty score). There was also a non-significant trend for gestational age (*p* = 0.08).

## 4. Discussion

In this large regional cohort of preterm-born children, 11.1% had behavioral problems at 5 years of age according to the SDQ, and 9.7% had borderline results. Behavioral troubles reported by the parents were highly correlated with sub-optimal school behavior as revealed by the GSA. Furthermore, a borderline score was also associated with school behavior difficulties; hence, the SDQ threshold in preterm children should be taken into account.

The percentage of SDQ abnormal and borderline scores in this preterm cohort did not differ from the general population of children in which the SDQ was tested [13] by Robert Goodman (11.1% vs. 10% abnormal scores and 9.7% vs. 10% borderline scores) [13]. Those results are comparable to those described in the EPIPAGE-2 cohort [1]: finding fewer behavioral disorders with increasing gestational age but a result roughly comparable to the reference population, particularly for preterm-born babies after 27 weeks. In our study, the male gender and intrauterine growth restriction resulted in an increased risk of a school and family behavior disorder. A review published in 2013 confirmed an increased risk of behavioral problems in children born prematurely and/or with low birth weight compared to children born at term at preschool ages (3 to 5 years old), with no statistically significant difference in the prematurity term [20]. Only one study included in this review did not show this relationship, from Baron et al., and concerned 3-year-old children and used the BASC-2 (Behavior assessment for Children, second edition) [21]. Note that this review only included children born before 2005. The evolution of developmental care and progress of neonatology in its reduction of cerebral aggression factors of systemic origin could have reduced the occurrence of behavioral problems at 5 years.

Risk factors associated with a significantly higher score of behavioral difficulties at 5 years in our cohort are male children and a low birth weight. This was also highlighted by Delobel-Ayoub et al. in 2009 [22]. Factors stemming from the babies were strongly associated with behavioral difficulties, but their medical explanations are still under discussion. For instance, recently, Giordano et al. found, in VLBW at the preschool age, an impact on the behavioral outcomes of neonatal morbidities, especially neonatal sepsis [23], whereas a recent review in 2016 did not emphasize specific perinatal risk factors, but found that behavioral outcome measures were heterogeneous [8].

A preterm behavioral phenotype has been proposed and widely reported, comprising difficulties with emotions (anxiety), attention and peer relationships [24]. Based on SDQ subscores, this pattern has been described in up to 20% of ELBW at 8 years old [25]. Academic achievement was associated with this pattern, especially mathematics and spelling performance. Only SDQ subscores have been assessed in this study and we demonstrate what an SDQ total score adds to the assessment of children’s behavior. We therefore propose, for the first time, to take into account the borderline SDQ total score threshold in preterm children.

Recently, in the EPIPAGE French cohort study about children born in 2011, there were no more behavioral troubles found with SDQ in the very preterm population than there were in a term-born control population at five years old [1]. Direct parent concern about children’s behavior is up to forty percent, asking what the relevant SDQ cut-off score is. Those results add to considering the borderline threshold rather than the abnormal threshold in the preterm population. Taking into account the delayed development of emotional regulation, the impact of age for the assessment of preterm children remains an issue. Concerning the evolution of the SDQ assessment profile with children’s age, Becker et al. [26] showed that in the general population of children aged 6 to 18 years, 6 years later, most children remain “normal” when the SDQ score classified them as such at the first assessment, whereas a small number of children remain “abnormal” or develop de novo disorders.

One of the strengths of our study is the high number of children who followed up in the network and were assessed at 5 years of age. From all included children in the neonatal period, 81% of the children were still followed up on at 5 years of age, and the parents of 77% correctly filled in the SDQ questionnaire and 68% filled in the two questionnaires correctly. Nevertheless, among the children not followed up at 5 years of age, the socioeconomic conditions of the parents were lower, with a greater number of parents receiving CMU. This point constituted a limit. We only observe a tendency between SDQ score and socioeconomic level but some studies have reported an association of behavioral problems and parents’ low socioeconomic level [12]. We then might have underestimated the occurrence of ex-preterm behavioral disorders. The cross-sectional study design constituted another limit of the study. Indeed, we are unable to predict, over a larger period of time, child behavior in the classroom with SDQ.

## 5. Conclusions

The behavior of preterm-born children was reported at 5 years of age by parents using an SDQ questionnaire, and it was found to be associated with the child’s behavior at school. The SDQ-related factors are the birthweight z-score and male sex, and stem from the child rather than the environment. Furthermore, an SDQ borderline score threshold has to be considered and should lead to an assessment of the child’s behavior.

## Figures and Tables

**Figure 1 children-10-01191-f001:**
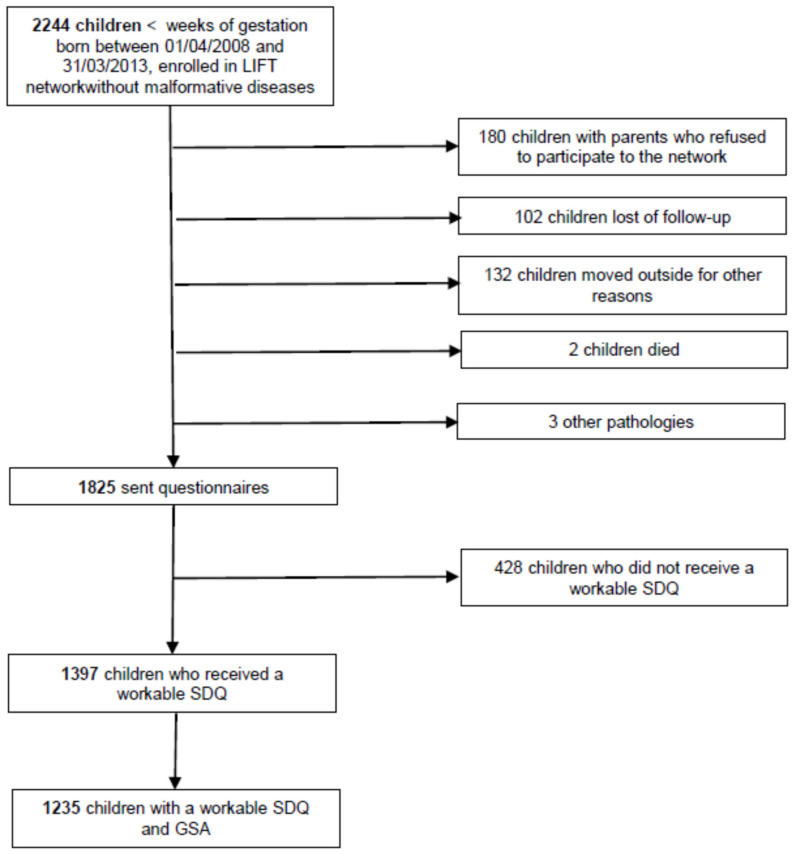
Flowchart.

**Figure 2 children-10-01191-f002:**
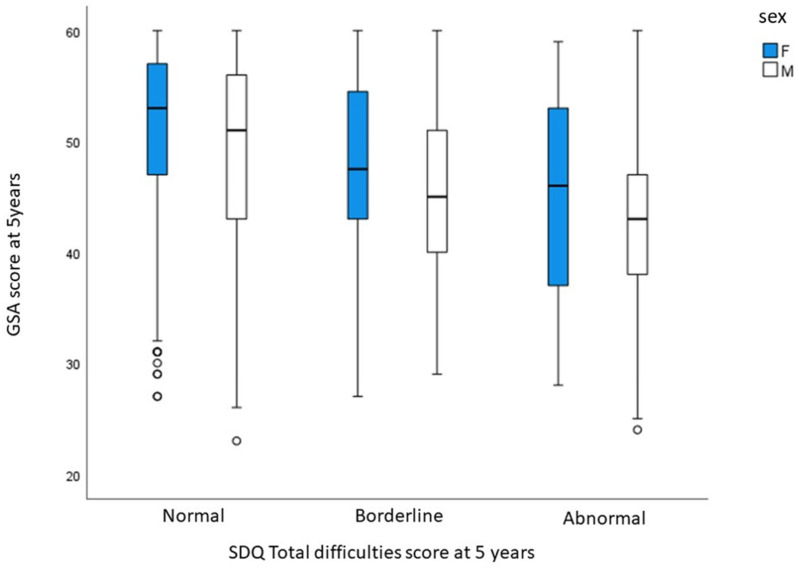
GSA score according to SDQ classification and sex.

**Table 1 children-10-01191-t001:** Characteristics of the population and comparison between the preterm children included and those not included.

	Included (n = 1397)	Not Included (n = 847)	*p*
Gestational age, wk (mean ± SD)	29.94 ± 2.00	29.79 ± 2.14	0.095
Birth weight Z-score * (median (IQR))	−0.18 (1.01)	−0.14 (1.06)	0.403
Female gender N (%)	641 (45.7%)	413 (48.9%)	0.186
Bronhcopulmonary dysplasia ** N (%)	82 (5.8%)	53 (6.2%)	0.583
Cerebral injury *** N (%)	42 (3.0%)	42 (5.0%)	0.06
CMUc N (%)	197(14.1%)	187 (22.1%)	<0.001
Breastfeeding at discharge N (%)	287 (20.5%)	187 (19.1%)	0.949

* The Z-scores were computed according to Olsen standards; ** Bronchopulmonary dysplasia defined by oxygenotherapy more than 28 days after birth. *** Cerebral injury defined by the presence of IVH III or IV, periventricular leukomalacia, or other cerebral anomaly on RMI at term.

**Table 2 children-10-01191-t002:** Total difficulty score by perinatal and socioeconomic data, results expressed as mean ± standard deviation or number (percentage %).

	Normal n = 1104 (79%)	Borderline n = 137 (9.8%)	Abnormal n = 156 (11.2%)	*p*
Gestational age, wk	29.99 ± 1.99	29.62 ± 2.14	29.87 ± 1.99	0.107
Birth weight Z-score	−0.15 (1.00)	−0.28 (1.05)	−0.36 (1.05)	0.032
Female gender	536 (48.6%)	46 (33.6%)	59 (37.8%)	<0.001
Bronchopulmonary dysplasia	63 (4.5%)	9 (6.5%)	10 (6.4%)	0.66
Cerebral injury	28 (2.5%)	5 (3.6%)	9 (5.8%)	0.256
CMUc	149 (13.5%)	21 (15.3%)	27 (17.3%)	0.059
Breastfeeding at discharge	222 (20.1%)	33 (24.1%)	32 (20.5%)	0.135

## Data Availability

Data Availability Statements are available in section “MDPI Research Data Policies” at https://www.mdpi.com/ethics.

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
