# Peer review of "Assessing Behavioral Disorders with SDQ in Very Preterm Children at 5 Years of Age in LIFT Cohort"

_children, 2023, doi:10.3390/children10071191_

Round 1

Reviewer 1 Report

The subject of the article refers to an important issue, which is the relationship between premature birth and the occurrence of behavioral disorders in children at an early stage of development. Accurate knowledge of these relationships is of great importance for the prevention of developmental disorders. Therefore, it is good that the authors of this article decided to explain this issue in a large group of 5-year-olds belonging to the LIFT preterm cohort children. The behavior of the surveyed children was assessed by teachers using the Global School Adaptation questionnaire and parents using the Strengths and Difficulties Questionnaire. Analysis of the results showed that in this relatively large group of 5-year-olds, 11.1% showed clear signs of behavioral disorders, while 9.7% had borderline results. As a reviewer, I positively assess the cognitive value of the article. Nevertheless, I would recommend expanding the "Conclusion" section, as it does not include all the important results of the study. In addition, the article requires a detailed linguistic correction (for example, on page 2, line 60 it is 'at 5 years old', and it should be 'at 5-year olds'). After making the suggested corrections, I support the publication of the article.

Author Response

Dear Reviewer,

Many thanks for your suggestions. In respect to your remark, manuscript has been improved and corrections has been made (such like “5-year olds”). Conclusion section has included important results to highlight what this study add.

Reviewer 2 Report

This study examined the cross-sectional associations between behavioral difficulties measured by the Strengths and Difficulties Parental Questionnaire (SDQ) and difficulty in global school adaptation measured by the Global School Adaptation Score (GSA) at 5 years in a group of preterm children.

This study had several flaws that limited its value.

1.     In abstract the authors described that “Questions remain about accuracy of self-reported parents questionnaires and real everyday life behavior functioning of the child.” It seemed that the authors planned to examined the validity of the SDQ in preterm children. However, the authors said that “the aim of this study was to evaluate behavioral difficulties reported by parents in preterm and behavioral school difficulties.” The aim of this study was not clear.

2.     The authors did not describe what new knowledge this study added to this filed. As the authors said, there have been several studies examining behavioral difficulties measured by other instruments. Maybe these previous studies have not examined the association between behavioral difficulties and global school adaptation, but the significant associations identified in this study seemed to be a matter of course.

3.     The data came from a longitudinal study. why did the authors examine the cross-sectional association between variables but not the prediction of behavioral difficulties for global school adaptation?

4.     The SDQ contained several subscales. Why did the authors use the total score but not the scores of the subscales?

5.     Writing is not good enough to make the readers understand. For example, “Unlike environment-related behavioral disorders, structural behavioral disorders in children are expressed in all these settings.

Need major revision.

Author Response

Dear reviewer,

We thank reviewer for relevant queries. Please find below a point by point response.

  1. In abstract the authors described that “Questions remain about accuracy of self-reported parents questionnaires and real everyday life behavior functioning of the child.” It seemed that the authors planned to examined the validity of the SDQ in preterm children. However, the authors said that “the aim of this study was to evaluate behavioral difficulties reported by parents in preterm and behavioral school difficulties.” The aim of this study was not clear.

We thank reviewer for this relevant remark, that allow us to improve presentation of the aim of this study. We propose line 12 : « To evaluate accuracy of SDQ validity reported by parents in preterm to predict behavioral difficulties in everyday school life environment »

  1. The authors did not describe what new knowledge this study added to this filed. As the authors said, there have been several studies examining behavioral difficulties measured by other instruments. Maybe these previous studies have not examined the association between behavioral difficulties and global school adaptation, but the significant associations identified in this study seemed to be a matter of course.

      That is a relevant suggestion. We add in the discussion what this study add in last sentence of SDQ publication datas paragraph. Line 197: Academic achievement is associated with this pattern, especially maths and spelling performance. Only SDQ subscores have been assessed in this study and we demonstrate what SDQ total score adds.

  1. The data came from a longitudinal study. why did the authors examine the cross-sectional association between variables but not the prediction of behavioral difficulties for global school adaptation?                              Thank you for this interesting suggestion. We have a brainstorming to discuss this possibility. We first want to publish association between SDQ and global school adaptation. With your remark, we will have a look about attrition at 7-year olds follow-up, for a future manuscript, to explore ability of SDQ to predict school adaptation. If so that would be an interesting tool for clinician. We thank reviewer for this suggestion.
  2. The SDQ contained several subscales. Why did the authors use the total score but not the scores of the subscales?                                                That is relevant point. We first want to explore threshold accuracy of SDQ score. Global score is used in research as well as in practice. Indeed, global score is available for GP, who usually have a look on result SDQ before child examination. But your remark is very relevant. We then introduce SDQ subscales data in discussion section. Line: Preterm behavioral phenotype has been proposed and widely reported, comprising difficulties with emotions (anxiety), attention and peer relationship ( Johnson S, Marlow N. Preterm birth and childhood psychiatric disorders. Pediatr Res. 2011). Based on SDQ subscores, this pattern has been described in up to 20% in ELBW at 8 years old (Exploring the “Preterm Behavioral Phenotype” in Children Born Extremely Preterm Alice C. Burnett). Academic achievement is associated with this pattern, especially maths and spelling performance. But SDQ total score has not been assessed in this study.

  1. Writing is not good enough to make the readers understand. For example, “Unlike environment-related behavioral disorders, structural behavioral disorders in children are expressed in all these settings.”                    Indeed, we removed this sentence and proposed a new version of the manuscript.

Reviewer 3 Report

It is necessary to explain the significance of this research and the urgency with which it must be conducted in the introduction.

The last section of the chart or flowchart needs to include information or data to explain why the number has decreased (1397 to 1235).

Please explain about research limitations.

Author Response

Dear reviewer,

Thank you very much for your relevant remarks that allow us to improve our manuscript.

It is necessary to explain the significance of this research and the urgency with which it must be conducted in the introduction.

RESPONSE: We thanks very much reviewer to highlight significance of this research. We propose in the introduction section : Child behavioral disorders can be related to family environment. These psychoaffective disorders are not observed in all lively place of the child, unlike constitutional disorders. To ensure that the SDQ parental reports explore constitutional disorders we compared SDQ to GSA.

The last section of the chart or flowchart needs to include information or data to explain why the number has decreased (1397 to 1235).

RESPONSE: Indeed, 1397 children have a SDQ at 5-year olds, but only 1235 have both SDQ and GSA questionnaire. For 162 children, GSA is missing.

We propose : A total of 2244 children were enrolled in the LIFT cohort during the study period, of whom 1825 (81,3%) had a visit at 5 years of age, 1397 (62,2%) were assessed with the SDQ and 1235 (55,0%) with both SDQ and GSA questionnaire

Please explain about research limitations.

RESPONSE: Indeed, we have not explain enough our main limitation. We propose: This point constituted a limit. We only observe a tendency between SDQ score and socio economic level but some studies have report an association of behavioral problems and low parents' socioeconomic level (13). We then might have underestimated occurrence of ex-preterm behavioral disorders.

Round 2

Reviewer 2 Report

The authors need to re-think what the aims of this study were and what knowledge this study brought to the world. The cross-sectional study design limited the value of this study.

The authors need to re-think what the aims of this study were and what knowledge this study brought to the world. The cross-sectional study design limited the value of this study.

Author Response

Dear reviewer, we are very grateful for your relevant comments. Indeed, we have to clearly state aim of the study. Please find here a point by point response,

COMMENT: The authors need to re-think what the aims of this study were and what knowledge this study brought to the world.

RESPONSE:

We propose into the abstract section line 12: Aim: To evaluate association between SDQ reported by parents in preterm and behavioral difficulties in everyday school life environment reported by teacher.

Into the introduction section, line 59:

To ensure that the SDQ parental reports explore constitutional disorders we compared SDQ to GSA. The objective of the present study is to investigate association between SDQ parental reports and child behavior in the classroom with GSA reported by teacher.

COMMENT:  What knowledge this study brought to the world

RESPONSE: We add to the discussion line 200: We therefore propose for the first time to take into account borderline SDQ total score threshold in preterm children.

COMMENT: The cross-sectional study design limited the value of this study.

RESPONSE: That is a relevant comment, we propose to add into the limit discussion section line 223:  Cross-sectional study design constituted another limit of the study. Indeed, we are unable to predict for along time child behavior in the classroom with SDQ.

We sincerely hope our response will reply to any comments. We have noticed that each comment enhanced our manuscript. We are very grateful for this.

Best regards

Round 3

Reviewer 2 Report

I maintain my original decision. Please stop torturing reviewers and authors. The study has been completed and the study limitations cannot be changed. So please do not ask authors to keep replying. It is a torture for authors.